# Development and Performance Evaluation of Cold-Patching Materials Using Waterborne Epoxy-Emulsified Asphalt Mixtures

**DOI:** 10.3390/ma13051224

**Published:** 2020-03-09

**Authors:** Yanqiu Bi, Rui Li, Sen Han, Jianzhong Pei, Jiupeng Zhang

**Affiliations:** School of Highway, Chang’an University, Xi’an 710064, China; biyanqiu@chd.edu.cn (Y.B.); lirui@chd.edu.cn (R.L.); hyram_hs@chd.edu.cn (S.H.); zhjiupeng@163.com (J.Z.)

**Keywords:** cold-patching materials, waterborne epoxy emulsified asphalt, waterborne epoxy mortar, initial strength, forming strength

## Abstract

Patching is one of the most common maintenance methods for potholes in roads. In order to improve the performance of cold-patching asphalt mixtures, an emulsified asphalt modified with waterborne epoxy resin was developed. Two waterborne epoxy resins and two curing agents were selected. The optimal experimental contents of the curing agents were obtained by measuring the compressive strength of the waterborne epoxy mortar (WEM) under different curing agent contents and curing period. The difference between the two waterborne epoxy resins was obtained by the flexural strength and stress–strain curves, which were measured by the modified bending test on the WEM. The evaluation method of the initial strength and forming strength of the waterborne epoxy emulsified asphalt mixture (WEEAM) was proposed by the experimental study of the compaction molding method and curing conditions. The high temperature performance, low temperature performance, and moisture susceptibility of the mixture were verified by comparing various kinds of WEEAM. The results show that using WEEAM as a road repair material has great advantages in improving pavement performance and road service levels.

## 1. Introduction

The rapid development of transportation in recent years has induced a significant increase in pavement distress. The development of road maintenance countermeasures is conducive to meeting economical, fast, and safe driving requirements [1]. Emulsified asphalt mixture (EAM) has gained extensive attention in research over the last few decades because of its superior workability at room temperatures and energy conservation. Wang Z. et al. [2] discussed the ranking regarding the effects of the binder’s content and curing age on the properties of EAM such as indirect tensile strength (IDT), dynamic stability (DS), IDT at low temperature, etc. The results indicate that emulsified asphalt content has a direct relation with IDT at low temperature. Nassar A. I. et al. [3] introduced a novel performance-based mix design method for EAM. The performance indicators included mechanical properties and volumetric properties. In addition, statistical methods were adopted to optimize the mix design parameters by using response surface methodology. M. Pasetto et al. [4] believed that a properly designed cold asphalt including asphalt shingles would be able to guarantee enhanced performance, demonstrating promising application as a patching repair for road maintenance activities.

However, EAM has not been widely applied as pavement materials due to its inadequate cohesiveness and low forming strength [5]. Many efforts have been made to solve these problems by utilizing the superior performance of polymers [6]. Among the modifiers of emulsified asphalt, waterborne epoxy has received a considerable attention for its excellent cementing performance [7]. Ding Q.J. et al. [8] investigated the applications of waterborne epoxy in cement emulsified asphalt concrete. It was found that waterborne epoxy improved the performance of concrete such as toughness, fatigue resistance, and crack resistance. Zeng D.L. [9] used waterborne epoxy-emulsified asphalt (WEEA) as the fog seal for road maintenance. The experiment results indicate that the waterproof effect of the pavement can be enhanced by adding waterborne epoxy.

Additionally, many attempts have been made to propose evaluation methodologies and indexes for the cold-patching materials of roads [10]. The initial strength and the forming strength were proposed to evaluate the performance of the cold patch asphalt mixture. The initial strength refers to the strength being just sufficient enough to resist the vehicle load after paving and compacting. Forming strength refers to the stable strength value obtained by the fully compacted mixture. There is no agreed requirement for the initial strength and forming strength of the cold-patching materials. The basic requirement for cold-patching materials in China suggests that initial strength should be greater than 3 KN and the forming strength should be greater than 5 KN by the Marshall method. The preparation methods of the initial strength and forming strength test specimens are different, and the test conditions are also different. The preparation of samples mainly considers the differences between compaction methods and curing methods. Curing temperature and curing time also varied from country to country, which affects the evaluation indicators of cold-patching materials. Sweden kept the mixture at a constant temperature environment of 60 °C for three days, while the U.S. research team simulated the formation strength of pavement at different curing times at room temperature. With regard to the test conditions, the samples may be damaged in the water bath or the value obtained is extremely low, according to the traditional Marshall stability test method, due to the low initial strength of the mixture. Therefore, it is recommended to measure the Marshall stability of the mixture as its initial strength without immersion at normal temperature conditions. Moreover, the Marshall test after the water is fully drained under high temperature curing conditions can be used to characterize its forming strength. As the molded mixture and the road surface jointly resist the load, the traditional Marshall test method can be used, that is, soaking in a constant temperature water bath at 60 °C ± 1°C for 30 min–40 min, and then measuring its Marshall stability to evaluate the forming strength of the mixture [11,12].

There have been many studies on cold-patching materials like those described above. However, most studies involve thermoplastic materials, so there is very little research on thermoset materials and there is a lack of uniform guidance. In order to compensate for the shortcomings of emulsified asphalt, waterborne epoxy resin with its strong bonding, fast forming speed, and high forming strength can be added. Considering the reinforcing of waterborne epoxy resin, it is worthwhile investigating the properties of WEEAM, especially in terms of the compaction molding method of WEEAM and an evaluation method of the strength and curing conditions. This paper presents the results of such a study.

## 2. Objectives

This paper presents a laboratory study to evaluate the properties of WEEAM at different curing temperatures and curing times. It is worth noting that the initial strength and forming strength of WEEAM for the performance evaluation of pavement pothole repair materials are proposed. In addition, the performance comparison of the two types of WEEAM with the emulsified asphalt mixture (EAM) and hot mix asphalt (HMA) was verified by laboratory experiments. 

## 3. Experimental Design

### 3.1. Materials

ESSO 90# asphalt binder and Jinyang emulsifier from Tianjin Province, China, were selected to prepare the emulsified asphalt. The properties of asphalt and prepared emulsified asphalt are shown in Table 1 and Table 2, respectively. Two types of waterborne epoxy resins were selected, named X waterborne epoxy and Y waterborne epoxy, respectively. Two types of curing agents were also selected, named A curing agent and B curing agent, respectively. The properties of the waterborne epoxy used in this work are displayed in Table 3. The limestone aggregate with a nominal maximum size of 9.5 mm and gradation shown in Figure 1 was produced from a local quarry in Shanxi Province, China, has a density of 2.702 g/cm^3^ and compressive strength of 135 MPa. 

### 3.2. Optimum Curing Agent Content

There is difference between the actual optimal ratio and the optimal stoichiometric ratio of the reactants, and this difference can lead to a reduced performance of the reaction product. The optimal reactant stoichiometric ratios were 3.33% for curing agent A and 39% for curing agent B, respectively. Five different percentages of curing agent A content (by weight of waterborne epoxy system), namely 1%, 2%, 3.33%, 4%, and 5% and five for curing agent B including 30%, 35%, 39%, 45%, and 50% were considered. 

Compressive strength is one of the parameters to evaluate the curing state of waterborne epoxy mortar (WEM) and the bonding performance of waterborne epoxy resin (WER). According to Chinese specification DL/T 5193-2004, the WEMs were cast into a cube specimen with the size of 40 mm [16]. In this study, we prepared specimens with curing ages of 3 d, 7 d, 14 d, and 28 d, respectively at a curing temperature of 23 ± 2 °C. The specimens were loaded with a continuous and uniform load using a universal testing machine (UTM) with a loading speed of 45 N/(mm^2^∙min). The maximum load when the sample is broken is recorded, and the compressive strength of the sample is obtained from the ratio of the maximum breaking load to the pressure-bearing area of the sample. Each test was repeated three times, and the compressive strength test results are shown in Figure 2.

From Figure 2a,b, it can be seen that the compressive strength of WEM increased with the increase in the curing agent content, and reached a peak at a certain content. Therefore, the actual optimal reactant ratios of the X waterborne epoxy system were 3.33% for curing agent A and 45% for curing agent B, respectively. Similarly, the actual optimal reactant ratios of the Y waterborne epoxy system were 3.33% for curing agent A and 45% for curing agent B, respectively, as can be seen in Figure 2c,d. Compared with X WEM, Y WEM presented a greater advantage in terms of compressive strength. However, it was observed that Y WEM was more brittle than X WEM during testing. Therefore, the selection of waterborne epoxy for pavement patch materials requires comprehensive consideration of the compressive strength and flexibility of the resin. 

The flexural strength test uses a revised bending test method based on the Chinese standard 0715-2011 [13]. The flexural strength test of WEM was conducted on Universal Testing Machine (UTM) to evaluate the difference in flexibility between the two waterborne epoxy resins. In order to obtain the product with the highest degree of curing, the samples were prepared at the actual optimal ratio of the waterborne epoxy and curing agent at 25 °C for 28 d. The mortars were prepared into beams with the size of 40 mm× 40 mm× 160 mm. The test was controlled at a displacement rate of 5 mm/min, and the test temperature was set to 25 °C. The flexural strength test results are displayed in Table 4 and Figure 3.

It can be observed from Table 5 and Figure 3 that the Y waterborne epoxy is more sensitive to the type of curing agent than the X waterborne epoxy. The ultimate flexural strength of Y WEM with curing agent A was similar to that of the X WEM with curing agent A. However, the ultimate flexural strength of Y WEM containing curing agent B increases significantly.

The honeycomb-like voids on the mortar surface and comminuted failure of the specimens were observed due to the poor compatibility of the Y waterborne epoxy with curing agent A. The failure load of WEM with curing agent B was three times or more than that of WEM with curing agent A. In addition, the displacement of X WEM was larger than that of Y WEM, which means that the flexibility of X WEM is generally better, and the failure of Y WEM is brittle fracture.

### 3.3. Evaluation Method of Initial Strength and Forming Strength

The proportion of waterborne epoxy emulsified asphalt (WEEA) to the mixture was determined to be 1:4, combining the existing experimental data of the mixture performance. The compaction molding method of WEEAM was designed by introducing a two-stage compaction method, according to the Chinese standard JTG F40-2004 [12]. The total number of compaction increased from 50 to 75, and the selected three types of two-stage compaction method are shown below: (i) compacted 25 times for the first stage, then curing at a constant temperature of 110 °C for 24 h, and compacted 25 times for the second stage; (ii) compacted 35 times for the first stage, then curing at a constant temperature of 110 °C for 24 h, and 40 times for the second stage; and (iii) compacted 50 times for the first stage, then curing at a constant temperature of 110 °C for 24 h, and compacted 25 times for the second stage. Specimens were prepared by using the methods described above and the indicators were measured, as shown in Table 5.

The Marshall test results presented in Table 5 indicate that the performance of WEEAM is closely related to the compaction molding method. The density of the samples increases and the air voids decrease with the increase in the compaction times. The stability of the mixtures was significantly improved with the increase in the number of compactions from 25 times to 35 times during the compaction of WEEAM. However, the aggregates were slightly broken and the emulsion splashed out of the molds when the number of compactions was increased from 35 to 50 in the first stage, which led to a lower density, higher air voids, and lower stability. Therefore, 35 times of compaction in the first stage, and 40 times of compaction in the second stage after curing were selected for further research. The compaction molding method of WEEAM is not only conductive to the formation of its strength, but also conforms to the compaction process of the cold-patching materials under load.

The preparation method of the modified Marshall was classified into room temperature curing and high temperature curing to conduct an initial strength test and forming strength test, respectively. The two-stage compaction method was proposed to conduct the test. Samples were compacted 35 times in the first stage and 40 times in the second stage after curing at 110 °C for 24 h, then curing at 25 °C for 48 h, 7 d, 14 d, and 28 d, respectively after demolding. The dehydration rate and the compressive strength of X WEEAM are given in Figure 4.

It can be observed from Figure 4 that the evaporation of water is slow at 25 °C, so the formation of strength is also relatively slow. Currently, the basic requirement of the initial strength is above 3 KN, so it is suitable to evaluate the initial strength of WEEAM by using the stability of the specimen after curing at 25 °C for 48 h.

The forming strength was evaluated by measuring the stability of the immersed specimen soaked in a 60 °C water bath for 30 min. Samples were compacted 35 times in the first stage and 40 times in the second stage after curing at 60 °C and 110 °C for 24 h, respectively, then curing at 25 °C for 48 h, 7 d, 14 d, and 28 d after demolding. The dehydration rate and compressive strength are shown in Figure 5.

From Figure 5, it can be clearly seen that the evaporation of water under the 60 °C curing was slower than that under 110 °C curing, and therefore the formation of the samples’ strength was also relatively slow. However, the specimen drained the water almost completely after curing at 110 °C for 48 h, and the final strength was almost formed. Hence, it is suitable for evaluating the forming strength of WEEAM by using the stability of the specimen after curing at 110 °C for 48 h.

### 3.4. Test Methods 

The wheel tracking test is widely utilized to simulate the application of an actual wheel load on pavement structures at high temperature. The modified two-stage rolling method was proposed to prepare rutting specimens according to the existing research results and the material characteristics in this research. First, the amount of material was calculated according to the density of the Marshall specimen, and then the uniformly mixed mixture was poured into a mold with the size of 300 mm × 300 mm × 50 mm according to Chinese specification T0719-2011 [13]. Then, the mixture in the mold was compacted back and forth seven times, and was then compacted seven times again after curing in a 110 °C thermostatic container for 24 h. Finally, the specimen was cured at room temperature for 24 h. A solid-rubber wheel travelling at a speed of 42 cycles/min and a wheel-pressure of 0.7 MPa were used to correlate with rutting. Two parameters, the DS and rutting depth at 60 min, were employed to feature high-temperature stability of the tested mixtures. Four specimens were tested for each group.

The low-temperature flexural test was employed to evaluate cracking resistance with a repeated load at low-temperature conditions according to Chinese standard 0715-2011 [13]. Asphalt mixtures were first fabricated by the wheel tracking device of the same dimension as the specimens for the wheel tracking test. Then, the specimens were cooled for 24 h, after which they were cut into beams with dimensions of 250 mm (length) × 30 mm (width) × 35 mm (height). The test was conducted at −10 °C with a loading rate of 50 mm/min. The maximum flexural tensile strain and stress at the bottom of the tested beams before failure were employed to characterize the anti-cracking performance of the tested mixtures at low temperature. Four specimens were tested for each group.

Moisture susceptibility is one of the most important cold patch asphalt mixture performances. The immersed Marshall test and freeze–thaw splitting test were carried out to evaluate the moisture susceptibility of WEEAM. Specimens were prepared by the method of the forming strength test. The immersed Marshall test is similar to the standard Marshall test, except that it was soaked in a 60 °C water bath for 48 h. Residual Marshall stability was used to evaluate the moisture susceptibility of WEEAM. In the freeze–thaw split test, four of these specimens were stored in room temperature under dry conditions. The other four specimens were processed following the procedures of Chinese T0729-2000, in which four specimens were subjected to continuous freezing at −18 °C for 16 h and thawing at 60 °C for 24 h. Then, the loading speed rate was 50 mm/min. Finally, the splitting test was carried out for all eight specimens after being immersed in 25 °C water bath for 2 h, according to Chinese standard T0715-2011 [13]. The parameter, freeze–thaw splitting tensile strength ration (TSR), was utilized to characterize the moisture susceptibility of the tested mixtures.

The Cantabro test was used to evaluate the raveling resistance of WEEAM [13]. The specimens were prepared by the method of the forming strength test. Four groups of specimens were prepared for each different type of mixture, and each group had three duplications. The specimen was put inside a Los Angeles Abrasion machine drum without steel balls, and the drum was turned to 300 revolutions at room temperature for 10 min. One specimen was tested at a time. The percentage of mass loss during the test was measured to evaluate the raveling resistance of different types of mixtures.

## 4. Results and Discussion

### 4.1. High Temperature Stability

High-temperature rutting is one of the most serious distresses of cold-patching materials [17]. Hence, the stability of different types of mixtures at high temperature were investigated, as shown in Figure 6.

Figure 6 shows the wheel tracking test results of the different types of mixtures. The dynamic stability of WEEAM increased by more than 98 times, and the rutting depth decreased by more than 11 times in comparison with EAM and HMA, indicating that waterborne epoxy has good high temperature rutting resistance. EAM is inferior in high temperature performance to HMA because of the presence of additives such as emulsifiers and stabilizers. HMA exhibits low viscosity and good rheology at high temperatures due to the thermoplasticity of the asphalt. However, the significant improvement in the high temperature performance of WEEAM is due to the thermosetting properties of the epoxy resin, which forms a three-dimensional continuous phase after curing [14,18]. Such mixtures are not only tough, but are also elastic at typical pavement service temperatures. The DS value of WEEAM could reach about 80–93% of the DS value of the epoxy asphalt mixture (EAM), and the rutting depths were comparable to the rutting depths of the epoxy asphalt mixture (EAM) [19,20]. 

### 4.2. Low Temperature Performance

Table 6 and Figure 7 present the low temperature crack resistance of different types of mixtures.

The results in Table 6 and Figure 7 indicate that there were significant differences in the low temperature crack resistance of different types of mixtures. It can be found that compared with the low temperature performance of HMA, the flexibility of WEEAM still needs to be improved. In fact, the flexural strength of WEEAM can reach 28–42% of HMA, and only 10–16% of the flexural strength of EMA [20]. This is mainly because the air voids of WEEAM are significantly larger than those of HMA due to the compaction molding method and material characteristics. Moreover, the closed voids formed by non-volatile water in WEEAM, which is the weakness of the materials, will lead to early failure of the specimen under the influence of external force. In addition, the epoxy resins were brittle at low temperatures and easily failed at lower strain, resulting in the poor fracture performance of WEEAM [18,20]. It can also be observed that the flexibility of Y WEEAM was worse than that of X WEEAM.

### 4.3. Moisture Susceptibility Analysis

Immersed Marshall test and freeze–thaw split test results are shown in Table 7 and Table 8, respectively.

From Table 7, it can be inferred that the residual Marshall stability reached more than 80%, which met the requirements of the moisture susceptibility standard of HMA, according to Chinese specification JTG F40-2004 [12]. The significant improvement in the moisture sensitivity of WEEAM is mainly due to the thermosetting properties and good bonding ability of epoxy resin. The epoxy resin forms a three-dimensional continuous phase, and the asphalt is dispersed therein after curing [18]. Therefore, it is difficult for water to corrode the binder on the surface of the aggregate. However, the closed voids formed by non-volatile water in WEEAM can cause its water sensitivity to be slightly worse than that of HMA.

According to Table 8, it can be found that waterborne epoxy had effects on the moisture susceptibility of the mixtures, likely due to the three dimensional networks in WEEAM formed by the thermoset epoxy resin. Although the TSR value of WEEAM did not reach 98% of the EAM, it was close to the TSR value of 76% of HMA [20]. Furthermore, the aggregates were wrapped in an interlocked binder formed by continuous epoxy and dispersed asphalt. The structure was far more stable, and it was difficult to destroy when the soggy mixture experienced successive freeze–thaw circles, indicating that WEEAM shows great resistance to freeze–thaw and moisture damage. However, the closed voids formed by non-volatile water in WEEAM can cause its TSR to be lower than that of HMA. This result is consistent with the results of the Immersed Marshall test.

### 4.4. Raveling Resistance Performance

The Cantabro test was conducted to evaluate the raveling resistance of different types of mixtures, and the results are listed in Table 9.

For raveling resistance, as can be seen from Table 9, the mass loss obviously decreased when the emulsified asphalt mixtures were mixed with waterborne epoxy. Mass loss of WEEAM decreased by over 66% in comparison with EAM without waterborne epoxy and was even lower than HMA, indicating that waterborne epoxy can increase the raveling resistance of the mixtures. WEEAM showed better cohesiveness than EAM because of the effect of the epoxy resin. Epoxy resins are a thermosetting material that possess high strength after curing. It is difficult to compact under vehicle load, thus, its large porosity will also facilitate the evaporation of water. Moreover, the raveling resistance of WEEAM was improved, also due to the strong bond between the aggregates and epoxy asphalt after the moisture evaporated almost completely.

## 5. Conclusions

Following the research trend in pavement maintenance on long-life and sustainable pavements that are more environmentally and user-friendly, a study was initiated to design WEEAM and evaluate its performance related to pavement functions. Through laboratory experiments, the following findings were obtained:(1)The type of curing agent has a significant effect on compression strength and flexural strength. Compared with the addition of curing agent A, the compression strength and flexural strength of WEM with curing agent B were significantly improved. In addition, the flexibility of X WEM was generally better, and the failure of Y WEM was brittle fracture.(2)The compaction molding method of WEEAM and the evaluation method of the initial strength and curing conditions were discussed. The use of a two-stage compaction Marshall method is recommended for testing. Samples for the initial strength test were compacted 35 times in the first stage and 40 times in the second stage after curing at 25 °C for 24 h, then curing at 25 °C for 48 h. It is suitable to evaluate the initial strength of WEEAM by using the stability. Samples for the forming strength test were compacted 35 times in the first stage and 40 times in the second stage after curing at 110 °C for 24 h, then curing at 25 °C for another 24 h. The forming strength was evaluated by measuring the stability of the immersed specimen soaked in a 60 °C water bath for 30 min.(3)The performance of WEEAM was verified by several laboratory experiments. The dynamic stability of WEEAM was increased by more than 98 times, and the rutting depth decreased by more than 11 times in comparison with EAM and HMA, indicating that waterborne epoxy has good high temperature rutting resistance. Compared with the low temperature performance of HMA, the flexibility of WEEAM still needs to be improved. The immersed Marshall test and freeze–thaw split test results suggest that WEEAM shows great resistance to freeze–thaw and moisture damage. For raveling resistance, the mass loss obviously decreased when emulsified asphalt mixtures were mixed with waterborne epoxy. Mass loss of WEEAM decreased by over 66% in comparison with EAM without waterborne epoxy and even lower than HMA, indicating that waterborne epoxy can increase the raveling resistance of mixtures.

## Figures and Tables

**Figure 1 materials-13-01224-f001:**
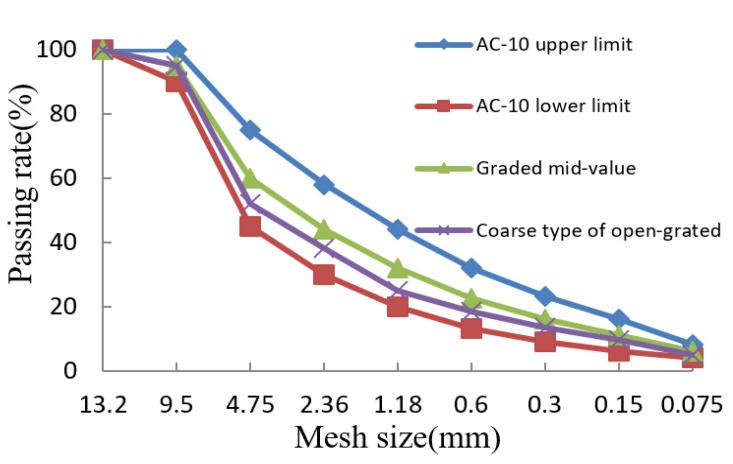
Asphalt mixture gradation curve of AC-10.

**Figure 2 materials-13-01224-f002:**
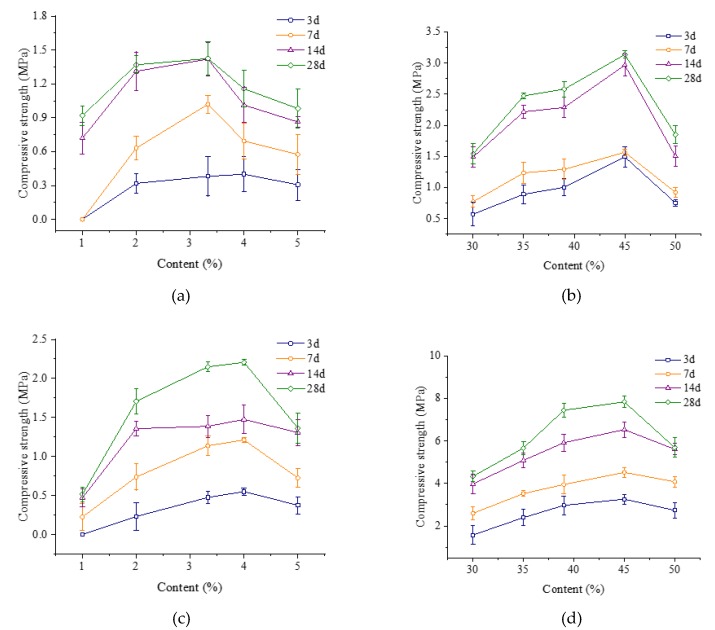
Compressive strength of different WEM with different curing agent and content. (**a**) Compressive strength of X WEM under different curing agent A contents; (**b**) compressive strength of X WEM under different curing agent B content; (**c**) compressive strength of Y WEM under different curing agent A content; (**d**) compressive strength of Y WEM under different curing agent B content.

**Figure 3 materials-13-01224-f003:**
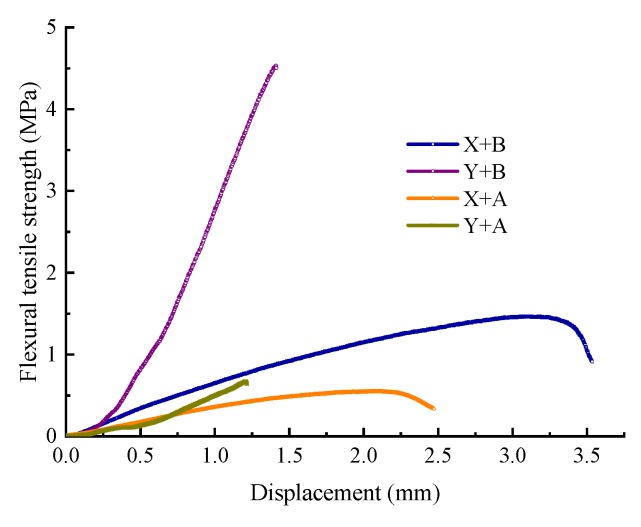
Flexural strength test results of WEM with different waterborne epoxies and curing agents.

**Figure 4 materials-13-01224-f004:**
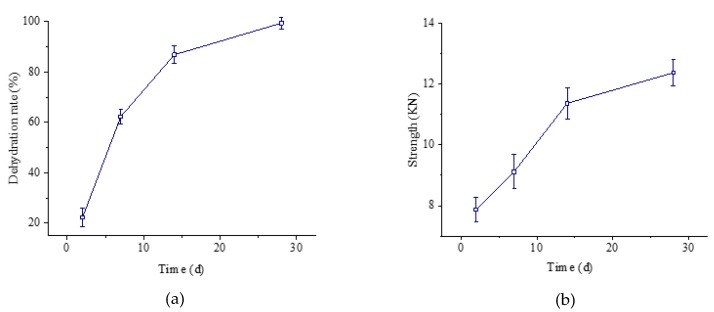
Dehydration rate and stability curve of WEEAM with different curing ages at 25 °C, (**a**) dehydration rate curve, (**b**) stability curve.

**Figure 5 materials-13-01224-f005:**
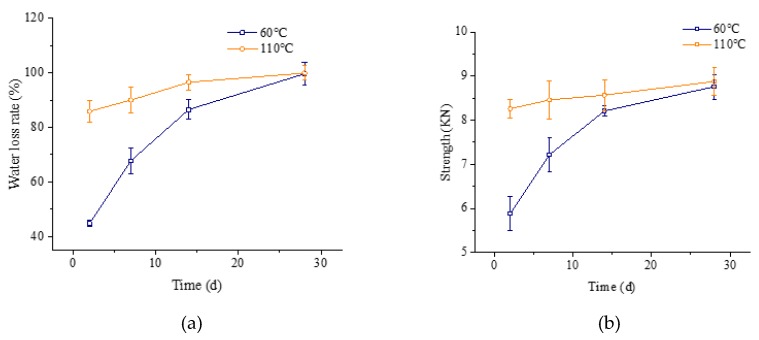
Dehydration rate and stability of WEEAM with different curing ages at high temperature. (**a**) Dehydration rate curve, (**b**) Stability curve.

**Figure 6 materials-13-01224-f006:**
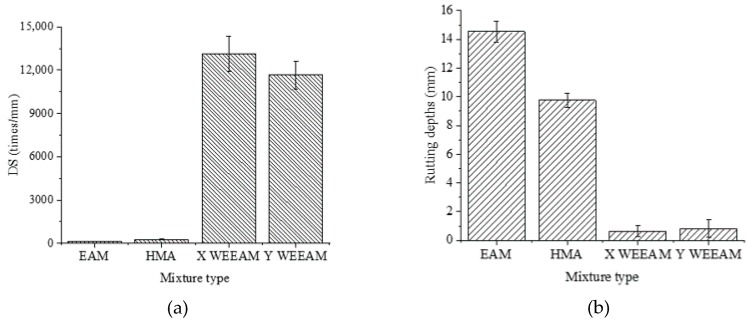
Wheel tracking test results of different types of mixtures. (**a**) dynamic stability, (**b**) rutting depths.

**Figure 7 materials-13-01224-f007:**
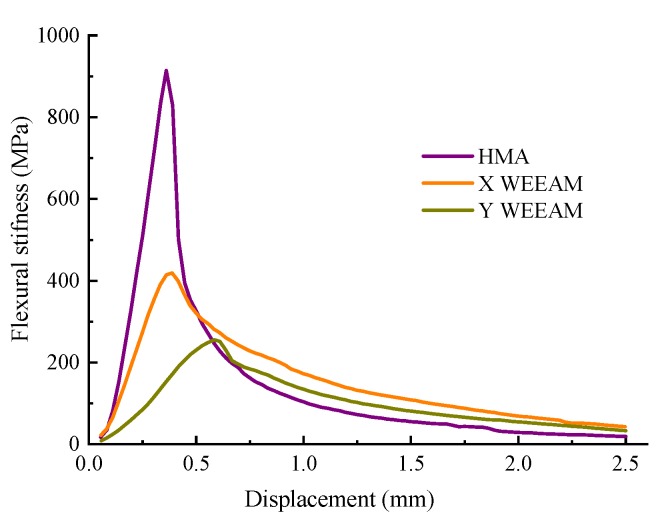
Low temperature crack resistance of different types of mixtures.

**Table 1 materials-13-01224-t001:** Properties of asphalt [13].

Indicators	Penetration (25 °C, 5 s, 100 g)/0.1 mm	Softening Point (R&B)/°C	Ductility (10 °C)/cm	Brookfield Viscosity (135 °C)/Pa∙S
Tested value	85.0	47.1	84.9	0.35
Standard value	80–100	>45	>40	-

**Table 2 materials-13-01224-t002:** Properties of emulsified asphalt [13].

Indicators	Sieve Residue (1.18 mm)/%	Standard Viscosity C_25.3_/s	Evaporation Residue			Storage Stability 25 °C, 5 d)/%
			Content of residual/%	Penetration (25 °C)/0.1 mm	Ductility (15 °C)/cm	
tested value	0.02	17.5	59.39	93.0	>100	3.2
standard value	≤0.1	10–60	≥55	45–150	≥40	≤5

**Table 3 materials-13-01224-t003:** Properties of the waterborne epoxy system [14,15].

Type	Solid Content/%	PH Value	Epoxy Value	Amine Hydrogen Equivalent
X waterborne epoxy	50 ± 2	6–8	0.137	-
Y waterborne epoxy	50 ± 2	6–8	0.137	-
A curing agent	100	-	-	-
B curing agent	50 ± 2	8–9	-	285

**Table 4 materials-13-01224-t004:** Flexural strength test results of different WEM with different curing agents.

Type	Failure Load P_B_ (KN)	Flexural Strength RB (MPa)	Mid-Span Deflection D(mm)	Bending Stiffness Modulus SB (MPa)	Maximum Bending Strain εB (με × 10^4^)
X + A	0.24	0.56	2.11	11	5.06
X + B	0.63	1.48	3.11	20	7.46
Y + A	0.29	0.68	1.21	23	2.90
Y + B	1.93	4.52	1.41	134	3.384

**Table 5 materials-13-01224-t005:** Marshall test results of WEEAM with different compaction molding method [13].

Compaction Method (Times)	Dry Density (g/cm^3^)	Air Void (%)	Asphalt Saturation (%)	Stability (KN)	Flow Value (0.1 mm)
25 + 25	2.33	9.2	67.5	8.26	24.93
35 + 40	2.35	8.5	68.6	9.32	22.33
50 + 25	2.34	8.7	67.9	8.44	23.57

**Table 6 materials-13-01224-t006:** Low temperature crack resistance test results of WEEAM.

Types of Mixtures	Failure Load P_B_ (KN)	Flexural Strength R_B_ (MPa)	Mid-Span Deflection D (mm)	Bending Stiffness Modulus S_B_ (MPa)	Maximum Bending Strain ε_B_ (με × 10^3^)
HMA	1.13	9.02	0.397	4291	2110
X WEEA	0.47	3.77	0.434	1728	2287
Y WEEA	0.31	2.56	0.556	883	2897

**Table 7 materials-13-01224-t007:** Immersion Marshall test results of different types of mixtures.

Mixture Type	Stability After Soaking for 48 h MS_2_ (KN)	Stability After Soaking for 30–40 min MS_1_ (KN)	Residual Marshall Stability MS_0_ (%)
HMA	6.12	7.52	81.4
EAM	2.69	4.32	62.2
X WEEAM	7.55	9.32	81.0
Y WEEAM	9.46	11.75	80.5

**Table 8 materials-13-01224-t008:** The freeze–thaw split test results of different types of mixtures.

Mixture Type	Tensile Strengths Under Freeze-Thaw Cycles RT2 (MPa)	Tensile Strengths Without Freeze-Thaw Cycles RT1 (MPa)	Tensile Strength Ratio TSR (%)
HMA	0.575	0.752	76.4
EAM	0.234	0.386	60.7
X WEEAM	0.654	0.859	76.1
Y WEEAM	0.736	0. 979	75.2

**Table 9 materials-13-01224-t009:** The Cantabro test results of different types of mixtures.

Mixture Type	Quality of Specimen Before Test m_0_ (g)	Quality of Specimen After Test m_1_ (g)	Mass Loss During Test Δs (%)
HMA	1220.47	1059.73	13.17
EAM	1235.85	793.42	35.80
X WEEAM	1214.30	1090.81	10.17
Y WEEAM	1230.49	1080.62	12.18

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
