# Peer review of "Development and Performance Evaluation of Cold-Patching Materials Using Waterborne Epoxy-Emulsified Asphalt Mixtures"

_materials, 2020, doi:10.3390/ma13051224_

Round 1

Reviewer 1 Report

The manuscrpt deals with the characterisation of cold patching materials using waterborne epoxy emulsified asphalt mixtures. The topic is not totally innovative, even if the approach and dissertation is sufficiently original for a publication in the journal.

A revision of the English language is strongly advised.

The term waterborne, in my opinion is not necessary; emulsions are usually made with water.

21. WEEMA has not been defined previously.

39 See also: Recycling bituminous shingles in cold mix asphalt for high-performance patching repair of road pavements (2017), Pavement and Asset Management, WCPAM 2017, Italy, DOI: 10.1201/9780429264702-75 CRC Press, p. 627-633, Taylor and Francis By M. Pasetto, G. Giacomello, E. Pasquini, A. Baliello.

51. initial strength and forming strength. Please clarify from which test.

Tables 1 to 3. Which standards have been considered?

93. The compressive strength should be described or the standard recalled.

105 and following. What is WME? It has not been defined.

116. Flexural strength tests according which standard? Which procedure was used (4PBT, 3PBT?) and why?

139. Please clarify the procedure. Standard?

Table 5 and 6 and 7. Please pay attention to the format (e.g. parentheses).

Table 5. Values of stability seem to be quite high for this type of material. The flow is quite low. Please confirm them.

146. The Marshall test was done after a curing period of 30 mins at a temperature of 60 C degrees in water?

196, 200, 209. Please clarify the standard.

258.  has meet the requirements of specifications in terms of moisture susceptibility of HMA... Which are the requirements?

279. decrease=decreases

Reviewer 2 Report

Generally, the text of the paper is well written. There are quite interesting research results.

The methods of research are solid. The presentation is technically correct. The interpretation of results is adequate.

Specific comments are as follows:

- In the manuscript Authors did not mention about statistical analysis. What statistical methods were used to statistical analysis in the experiment? e.g. ANOVA, logistic regression, etc. (Table 4, 5, 6 and others). In Table 6 is Mpa instead of MPa.

- Fig 2. - please provide error bars for the compressive strength.

- Figs. 4 and 5 - please provide error bars for the dehydration rate and stability.

In the remaining charts, please also complete.
All error bars should be accompained by their meanings: e.g. value +/-
SD (SD-standard deviation).
- Table 5 - Please organize brackets.

- Authors need to provide comparative studies with similar research reported in the literature. 

Generally, authors should refer on relevant tables and figures when discussing their results. They should pay more attention to comparison of their results with similar publications, which will be correct, and the referring on their data (Results section) would be mostly recommended. Hence the small amount of references.

This part of the article is important to present:

- the background and context

- related studies and actual knowledge

- why this study is pertinent and how could potentially improve the knowledge.

- References are not prepared in accordance with the mdpi guidelines, there are also no doi numbers.

According my suggestion the paper needs restructuration and complete discussion.
I recommend the paper to publish after major revisions.

Round 2

Reviewer 2 Report

The authors meticulously approached the reviewer's comments. Many parts of the article have been corrected.

Minor errors appear, e.g. in Table 6. Flexural strength unit is in Mpa, please change to MPa. Bending stiffness modulus too.

I accept the paper for publication.